# Studies of the Anti-Diabetic Mechanism of *Pueraria lobata* Based on Metabolomics and Network Pharmacology

**Shu Zhang** [1], **Qi Ge** [2,3], **Liang Chen** [3] **and Keping Chen** [3,*]

1 School of Food and Biological Engineering, Jiangsu University, Zhenjiang 212013, China; njzs1987@sina.com
2 School the Environment and Safety Engineering, Jiangsu University, Zhenjiang 212013, China
3 School of Life Sciences, Jiangsu University, Zhenjiang 212013, China; geqi0616@163.com (Q.G.); oochen@ujs.edu.cn (L.C.)
* Correspondence: kpchen@ujs.edu.cn

**Abstract:** Diabetes mellitus (DM), as a chronic disease caused by insulin deficiency or using obstacles, is gradually becoming a principal worldwide health problem. *Pueraria lobata* is one of the traditional Chinese medicinal and edible plants, playing roles in improving the cardiovascular system, lowering blood sugar, anti-inflammation, anti-oxidation, and so on. Studies on the hypoglycemic effects of *Pueraria lobata* were also frequently reported. To determine the active ingredients and related targets of *Pueraria lobata* for DM, 256 metabolites were identified by LC/MS non targeted metabonomics, and 19 active ingredients interacting with 51 DM-related targets were screened. The results showed that puerarin, quercetin, genistein, daidzein, and other active ingredients in *Pueraria lobata* could participate in the AGE-RAGE signaling pathway, insulin resistance, HIF-1 signaling pathway, FoxO signaling pathway, and MAPK signaling pathway by acting on VEGFA, INS, INSR, IL-6, TNF and AKT1, and may regulate type 2 diabetes, inflammation, atherosis and diabetes complications, such as diabetic retinopathy, diabetic nephropathy, and diabetic cardiomyopathy.

**Keywords:** *Pueraria lobata*; anti-diabetic; network pharmacology; active ingredients





## 1. Introduction

Diabetes mellitus (DM), including type 1, type 2, and gestational diabetes, is a chronic and metabolic disease with long-term hyperglycemia and complex pathogenesis, influenced by interactions of multiple factors such as genetics, environment, age, and sex. According to the official website of the International Diabetes Federation, the global prevalence of diabetes among adults aged 20–79 years was 463 million, which means one out of eleven had diabetes in 2019, and that number is expected to rise to 578 million by 2030. Seriously, China had been the top country for the number of adults with diabetes (data from: IDF Diabetes Atlas Globally, 9th ed., 2019). The sixth large-scale diabetes epidemiological survey in China showed a trend of gradual increase in prevalence rates from 2007 to 2017, reaching 12.8% in 2017. The proportion of pre-diabetes was as high as 35.2%, and nearly half of Chinese people had abnormal blood sugar levels. No matter the therapeutic methods, the propaganda on the prevention of diabetes leaves a lot to be desired [1]. Therefore, traditional Chinese medicine (TCM) with low cost and stable efficacy has become a complementary and alternative medicine for treating DM [2,3].

Recently, old-line natural plants which are defined as medicine and food, sharing the same origin, have gradually played a significant role in medical care due to low toxicity and side effects, wide material access, and perfect curative effects. The dried roots of *Leguminosae Kudzu*, and *Pueraria lobata* are widely distributed in China and approved as both food and medicine on the list by the National Health Bureau [4,5]. According to the Dictionary of Chinese Medicine, *Pueraria lobata* and *Pueraria thomsonii* are the sources of authentic medicinal products. *Pueraria lobata* is sweet and pungent in taste and cool in nature [6]. It has been documented in Shen Nong Ben Cao Jing (206 BC–24 AD, Western

Dynasty) and has the functions of relieving fever, generating body fluid, and treating DM, classified as Xiao Ke syndrome in TCM. Contemporary pharmacological research has indicated that the main ingredients such as isoflavones, steroids, and terpenoids have positive effects on analgesia, regulating blood pressure and sugar, anti-alcoholism, anti-tumor, and so on. Research shows that isoflavones in the extracts of Pueraria as puerarin can reduce blood glucose in STZ induced diabetic mice with the expression of insulin [7]. Besides, puerarin may be used to treat diabetic nephropathy, a common complication of diabetes, by inhibiting a majority of growth factors, blocking protein glycosylation, and reducing the accumulation of extracellular matrix (ECM) in the kidneys [8–10]. Besides, studies on the main ingredients and pharmacology of *Pueraria lobata* have been conducted for a certain period, but the underlying mechanism of the active ingredients interfering with diabetes has not yet been clarified.

Compared with traditional methods, network pharmacology, including molecular dynamics, docking technology, and data analysis via computer, has been widely applied in the investigations of natural products with the characteristic of being multi-targeting for the treatment of human diseases [11]. In this present study, we attempted to screen the active ingredients and constructed the compound–target–disease networks through the system pharmacology and LC/MS metabolomics, to infer the potential mechanism of *Pueraria lobata* for treating DM. Detailed procedures of this study can be seen in Figure 1.

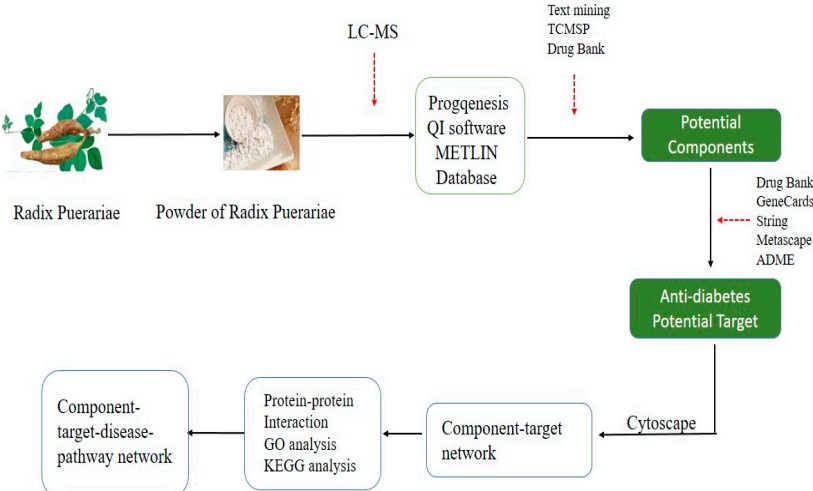

**Figure 1.** The flowchart of the network pharmacological analysis approach.

## 2. Materials and Methods

### 2.1. Raw Materials of Pueraria lobata Preparing

The roots of *Pueraria lobata* were purchased from Bozhou, Anhui Province, China, in 2020, and the roots of the healthy plants picked in autumn were washed, dried, and ground to a fine powder in an electric grinder in the School of Life Sciences, Jiangsu University, Jiangsu Province, China. The 30 mg accurately weighted *Pueraria lobata* powder was transferred to a 1.5 mL Eppendorf tube. Two small steel balls were added to the tube, 20 µL internal standard (0.3 mg/mL, 2-chloro-L-phenylalanine, dissolved in methanol) and 1 mL extraction solvent with methanol/water (7/3, *v/v*) were added to the tube, and then stored at −80 °C for 2 min, ground at 60 Hz for 2 min, ultrasonicated for 30 min, and finally placed at −20 °C for 20 min. Samples were centrifuged at 13,000 rpm, at 4 °C for 10 min. The supernatant of 300 µL was dried and then redissolved with 400 µL methanol/water (1/4, *v/v*), and then vortexed for 30 s and stored at 4 °C for 2 min. The extract was centrifuged at 13,000 rpm, 4 °C for 10 min. 150 µL supernatants were collected using crystal syringes, filtered through 0.22 µm microfilters, and transferred to LC vials. The rest of the vials were stored at −80 °C until LC/MS analysis, and all chemicals and reagents were analytical or HPLC grade (gradient grade).

## 2.2. Liquid Chromatography-Mass Spectrometry Experiments

The derivatized samples were analyzed on a Dionex Ultimate 3000 RS UHPLC system fitted with a Q-Exactive quadrupole-Orbitrap mass spectrometer, and equipped with a heated electrospray ionization (ESI) source (Thermo Fisher Scientific, Waltham, MA, USA). An ACQUITY UPLC BEH C18 column (2.1 × 100 mm, 1.7 μm) was employed in both positive and negative modes. The binary gradient elution consisted of (A) water (containing 0.1% formic acid, *v/v*) and (B) methanol separated at the following gradient: 0.01 min, 5% B; 1.5 min, 5% B; 3 min, 30% B; 7 min, 60% B; 9 min, 90% B; 11 min, 100% B; 12 min, 100% B; 12.1 min, 5% B; and 15 min, 5% B. The injection volume was 10 μL. The flow rate was 0.35 mL/min and the column temperature was 45 °C. The mass range was from Da 60 to 900. The resolution was set at 70,000 for the full scans, and 17,500 for HCD MS/MS scans. The mass spectrometer operated as follows: spray voltage, 3500 V (+) and 3100 V (−); sheath gas flow rate, 30 arbitrary units; auxiliary gas flow rate, 10 arbitrary units; capillary temperature, 320 °C.

## 2.3. Chemical Ingredients Determination and Active Compounds Screening

The acquired raw data were preliminarily treated by the progenesis Q I v2.3 software (Waters Corporation, Milford, CT, USA) based on the following parameters: precursor tolerance 5 ppm; product tolerance 10 ppm; product ion threshold 5%. Metabolites were identified by the public databases such as METLIN (http://metlin.scripps.edu/index.php, accessed on 4 March 2020). The chemical ingredients obtained from LC/MS were input into some public databases such as TCMSP (http://lsp.nwsuaf.edu.cn/tcmsp.php, accessed on 4 March 2020), DrugBank (https://go.drugbank.com/, accessed on 4 March 2020), GeneCards (https://www.genecards.org/, accessed on 4 March 2020), and Metascape (https://metascape.org/, accessed on 4 March 2020), to screen the active ingredients that interact with DM [12]. The ADME parameter-based virtual screening of the ingredients was utilized to further identify anti-diabetic ingredients using oral bioavailability (OB > 30%) and drug-likeness (DL > 0.18) as parameters [13,14].

## 2.4. Targets Prediction and Protein-Protein Interaction Network Construction

To identify the corresponding DM-targets of the active compounds of *Pueraria lobata*, several approaches were used. First of all, the target proteins of the active ingredients were selected, and duplicates removed based on TCSMP and Drugbank. Then, targets for DM using RS (relevance score > 28) as specific criteria were screened with the DrugBank Database. Finally, the common targets, both in ingredients and DM, were uploaded to the STRING (http://string-db.org, accessed on 4 March 2020) online website to obtain the information of protein–protein interactions (PPI). We selected confidence data > 0.7 to ensure the reliability of the analysis. The acquired data were analyzed with Cytoscape 3.8.2 software (https://cytoscape.org/, accessed on 18 July 2021).

## 2.5. Gene Ontology (GO) and KEGG Enrichment Analysis of Targets

The Database for Annotation, Visualization, and Integrated Discovery (DAVID, https://david.ncifcrf.gov/, accessed on 4 March 2020) v6.8 was used to analyze the GO function and KEGG pathway enrichment of proteins.

## 3. Network Construction

After obtaining the interaction information of the active ingredient compounds, target proteins, and the pathways, the "Ingredient–Target" and "Drug–Ingredient–Target Pathway" network was established by Cytoscape 3.8.2 software. Then, a hypothesized schematic diagram of the target proteins involved in the pathways was drawn.

## 4. Results

### 4.1. Chemical Metabolites of Pueraria lobata by LC/MS

To ensure the accuracy of the results, the sample of *Pueraria lobata* was analyzed three times by LC/MS. The results showed strong signals, high peak capacity, and high reproducibility. Two-hundred and fifty-six chemical metabolites were detected, containing phenylpropanoids and polyketides such as formononetin, ononin, genistein, acacetin, and benzenoids such as emodin, organic acids and derivatives, lipids and lipid-like molecules, and so on. The base peak chromatogram of positive and negative ions and the composition of the chemical metabolites are shown in the Supplementary Material Figure S1. The specific information of the 256 chemical metabolites is set out in Supplementary Material Table S1.

### 4.2. Active Ingredients from Pueraria lobata

Assessment of the ADME properties of active ingredients has become an important process in modern drug discovery. In the present work, two ADME, including OB and DL, were employed to screen the active ingredients of *Pueraria lobata.* Therefore, 12 out of 18 ingredients passed through the strict criteria: OB (>30%) and DL (>0.18), and most of them exhibited strong pharmacological activities. For instance, formononetin displayed significant hypoglycemic activity [15]; beta-sitosterol showed therapeutic effects on DM, anti-oxidation, and reducing blood lipids [16], and quercetin played an important role in treating obesity and DM [17,18]. In addition, the ononin, genistein, daidzein, daidzin, and tangeritin were detected in the metabolites of *Pueraria lobata* by LC/MS, and their certain pharmacological effects in anti-inflammatory, anti-diabetic and anti-tumor aspects were reported [19–21]. For example, ononin displayed effects on LPS-induced inflammatory response, which may be associated with NF-κB and MAPK pathways [22]; genistein showed the pharmacological effects on breast tumor, prostate tumor, atherosclerosis, and DM [23–25]. Surprisingly, puerarin, the main active and characteristic marker components for *Pueraria lobata* in Chinese Pharmacopoeia (The State Pharmacopoeia Commission of China, 2020), exhibited low OB and DL. Nevertheless, puerarin had lipid- and glucose-lowering effects in the treatment of obesity and DM [26–28], together with anti-inflammation effects, and the improvement of cardiovascular problems [29–31]. In summary, it was reasonable to believe that the 19 differentiated compounds could be listed as potential effective pharmacological activities for *Pueraria lobata* (Table 1).

**Table 1.** Information of the active ingredients of *Pueraria lobata*.

| MOL ID | Name | CAS No. | Molecular Formula | OB% | DL | Structure |
|---|---|---|---|---|---|---|
| MOL001689 | Acacetin | 480-44-4 | $C_{16}H_{12}O_5$ | 34.97 | 0.24 |  |
| MOL005814 | Tangeritin | 481-53-8 | $C_{20}H_{20}O_7$ | 21.38 | 0.43 |  |
| MOL002881 | Diosmetin | 520-34-3 | $C_{16}H_{12}O_6$ | 31.14 | 0.27 |  |
| MOL011787 | Glycitin | 40246-10-4 | $C_{22}H_{22}O_{10}$ | 22.48 | 0.78 |  |

**Table 1.** *Cont.*

| MOL ID | Name | CAS No. | Molecular Formula | OB% | DL | Structure |
|---|---|---|---|---|---|---|
| MOL004328 | naringenin | 480-41-1 | $C_{15}H_{12}O_5$ | 59.29 | 0.21 | |
| MOL005828 | Nobiletin | 478-01-3 | $C_{21}H_{22}O_8$ | 61.67 | 0.52 | |
| MOL000098 | Quercetin | 117-39-5 | $C_{15}H_{10}O_7$ | 46.43 | 0.28 | |
| MOL002714 | Baicalein | 491-67-8 | $C_{15}H_{10}O_5$ | 33.52 | 0.21 | |
| MOL000472 | Emodin | 518-82-1 | $C_{15}H_{10}O_5$ | 24.4 | 0.24 | |
| MOL000392 | formononetin | 485-72-3 | $C_{16}H_{12}O_4$ | 69.67 | 0.21 | |
| MOL000391 | Ononin | 486-62-4 | $C_{22}H_{22}O_9$ | 11.52 | 0.78 | |
| MOL000390 | Daidzein | 486-66-8 | $C_{15}H_{10}O_4$ | 19.44 | 0.19 | |
| MOL000481 | Genistein | 446-72-0 | $C_{15}H_{10}O_5$ | 17.93 | 0.21 | |
| MOL000358 | beta-sitosterol | 83-46-5 | $C_{29}H_{50}O$ | 36.91 | 0.75 | |
| MOL002959 | 3′-Methoxydaidzein | 21913-98-4 | $C_{16}H_{12}O_5$ | 48.57 | 0.24 | |
| MOL003629 | Daidzein-4,7-diglucoside | 53681-67-7 | $C_{27}H_{30}O_{14}$ | 47.27 | 0.67 | |
| MOL012297 | puerarin | 3681-99-0 | $C_{21}H_{20}O_9$ | 24.03 | 0.69 | |

**Table 1.** *Cont.*

| MOL ID | Name | CAS No. | Molecular Formula | OB% | DL | Structure |
|---|---|---|---|---|---|---|
| MOL004631 | 7,8,4′-Trihydroxyisoflavone | 75187-63-2 | $C_{15}H_{10}O_5$ | 20.67 | 0.22 | |
| MOL009720 | daidzin | 552-66-9 | $C_{21}H_{20}O_9$ | 14.32 | 0.73 | |

### 4.3. Target Proteins of Pueraria lobata

Experimental approaches which are usually used for searching the targets of drugs are considered time-consuming and as overspending. In this work, a macroscopic approach was applied to identify the target proteins for the effective ingredients of *Pueraria lobata*. A total of 318 targets of 19 effective ingredients of *Pueraria lobata* were searched by using TCMSP. Meanwhile, 219 target proteins for DM which passed through specific criteria, RS (relevance score > 28), were screened based on the DrugBank Database. Finally, 51 DM-related targets common to both targets of 19 effective ingredients and DM were determined (Table 2). The PPI network construction of these targets and their interactions were demonstrated by using STRING (Figure 2). The results included 51 nodes and 309 edges, of which nodes represented the target proteins and the edges represented the interactions between the proteins. Furthermore, the larger the degree was, the stronger the relationship between the proteins corresponding to the node in this network. Thus, it was indicated that the target proteins played a key role in the whole interaction network, which is the important target protein. IL6, AKT1, VEGFA, TNF, FN1, STAT3, TP53, CCl2, and IL1B were the top 10 proteins with degree values in the protein network interaction map. The statistical bar chart of proteins with a degree value greater than 12 is shown in Supplementary Material Figure S2.

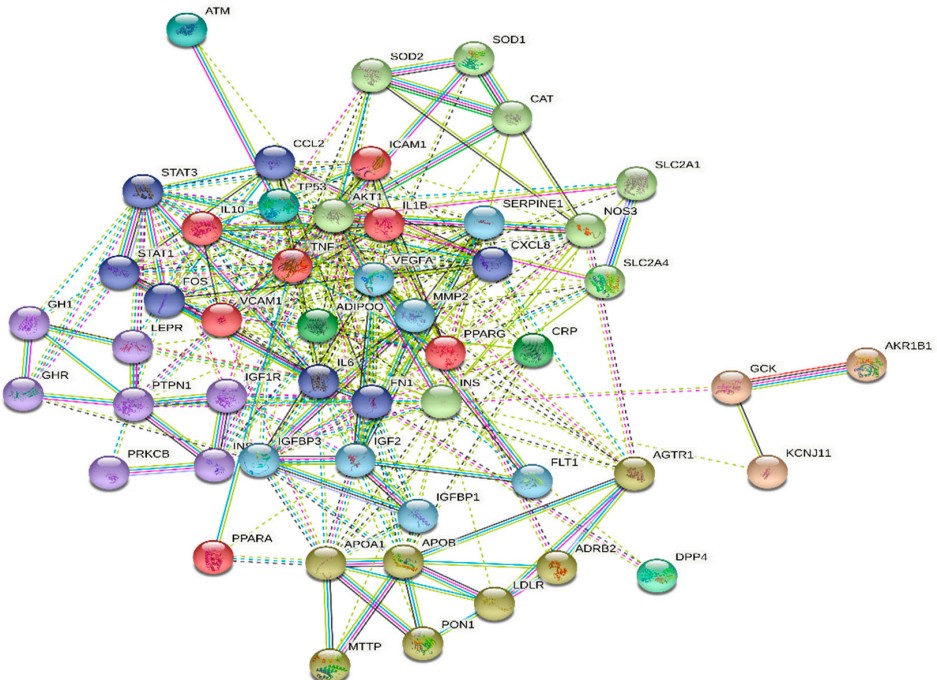

**Figure 2.** Target protein interaction (PPI) network analysis.

**Table 2.** Information of the DM-targets of *Pueraria lobata*.

| Uniport | Gene | Target Protein |
|---|---|---|
| P27487 | DPP4 | Dipeptidyl Peptidase 4 |
| P07550 | ADRB2 | Adrenoceptor Beta 2 |
| P04637 | TP53 | Cellular tumor antigen p53 |
| P29474 | NOS3 | Nitric Oxide Synthase 3 |
| P31749 | AKT1 | AKT Serine/Threonine Kinase 1 |
| P01130 | LDLR | Low Density Lipoprotein Receptor |
| P00441 | SOD1 | Superoxide Dismutase 1 |
| P04040 | CAT | Catalase |
| P37231 | PPARG | Peroxisome Proliferator Activated Receptor Gamma |
| P55157 | MTTP | Microsomal Triglyceride Transfer Protein |
| P04114 | APOB | Apolipoprotein B |
| Q07869 | PPARA | Peroxisome Proliferator Activated Receptor Alpha |
| Q15848 | ADIPOQ | Adiponectin, C1Q And Collagen Domain Containing |
| P18031 | PTPN1 | Protein Tyrosine Phosphatase Non-Receptor Type 1 |
| P15121 | AKR1B1 | Aldo-Keto Reductase Family 1 Member B |
| P15692 | VEGFA | Vascular Endothelial Growth Factor A |
| P01100 | FOS | Proto-oncogene c-Fos |
| P08253 | MMP2 | Matrix Metallopeptidase 2 |
| P22301 | IL10 | Interleukin 10 |
| P01375 | TNF | Tumor Necrosis Factor |
| P05231 | IL6 | Interleukin 6 |
| P42224 | STAT1 | Signal Transducer and Activator Of Transcription 1 |
| P05362 | ICAM1 | Intercellular Adhesion Molecule 1 |
| P01584 | IL1B | Interleukin 1 Beta |
| P13500 | CCL2 | C-C Motif Chemokine Ligand 2 |
| P19320 | VCAM1 | Vascular Cell Adhesion Molecule 1 |
| P10145 | CXCL8 | C-X-C Motif Chemokine Ligand 8 |
| P05771 | PRKCB | Protein Kinase C Beta |
| P05121 | SERPINE1 | Plasminogen activator inhibitor 1 |
| P14672 | SLC2A4 | Solute Carrier Family 2 Member 4 |
| P06213 | INSR | Insulin Receptor |
| P02741 | CRP | C-Reactive Protein |
| P17936 | IGFBP3 | Insulin Like Growth Factor Binding Protein 3 |
| P01344 | IGF2 | Insulin Like Growth Factor 2 |
| P27169 | PON1 | Paraoxonase 1 |
| P17948 | FLT1 | Fms Related Receptor Tyrosine Kinase 1 |
| P11166 | SLC2A1 | Solute Carrier Family 2 Member 1 |
| P08069 | IGF1R | Insulin Like Growth Factor 1 Receptor |
| P04179 | SOD2 | Superoxide Dismutase 2 |
| P01241 | GH1 | Growth Hormone 1 |
| P08833 | IGFBP1 | Insulin Like Growth Factor Binding Protein 1 |
| P10912 | GHR | Growth Hormone Receptor |
| P40763 | STAT3 | Signal Transducer and Activator of Transcription 3 |
| P02751 | FN1 | Fibronectin 1 |
| Q13315 | ATM | ATM Serine/Threonine Kinase |
| P01308 | INS | Insulin |
| P02647 | APOA1 | Apolipoprotein A1 |
| P35557 | GCK | Glucokinase |
| Q14654 | KCNJ11 | ATP-sensitive inward rectifier potassium channel 11 |
| P30556 | AGTR1 | Angiotensin II Receptor Type 1 |
| P48357 | LEPR | Leptin Receptor |

*4.4. Go Enrichment Analysis and KEGG Classification of Target Proteins*

Analysis of interaction network regulation of 51 targets was performed by DAVID. The top 15 significantly enriched terms in Cellular Component, Biological Process, and Molecular Function categories ($p < 0.05$, *p*-values were corrected using the Benjamin–Hochberg procedure) were listed in Figure 3. It was interesting to find these huge quantities of targets that interacted with a variety of Biological Process terms such as response to

peptide, response to insulin, regulation of small molecule metabolic process, and blood vessel morphogenesis. In the Molecular Function category, the target proteins were mainly involved in receptor-ligand activity, phosphatase binding, protease binding, and insulin-like growth factor binding, while in the Cellular Component category, the target proteins were classified into receptor complex, endoplasmic reticulum lumen, cytoplasmic vesicle lumen, and plasma lipoprotein particles.

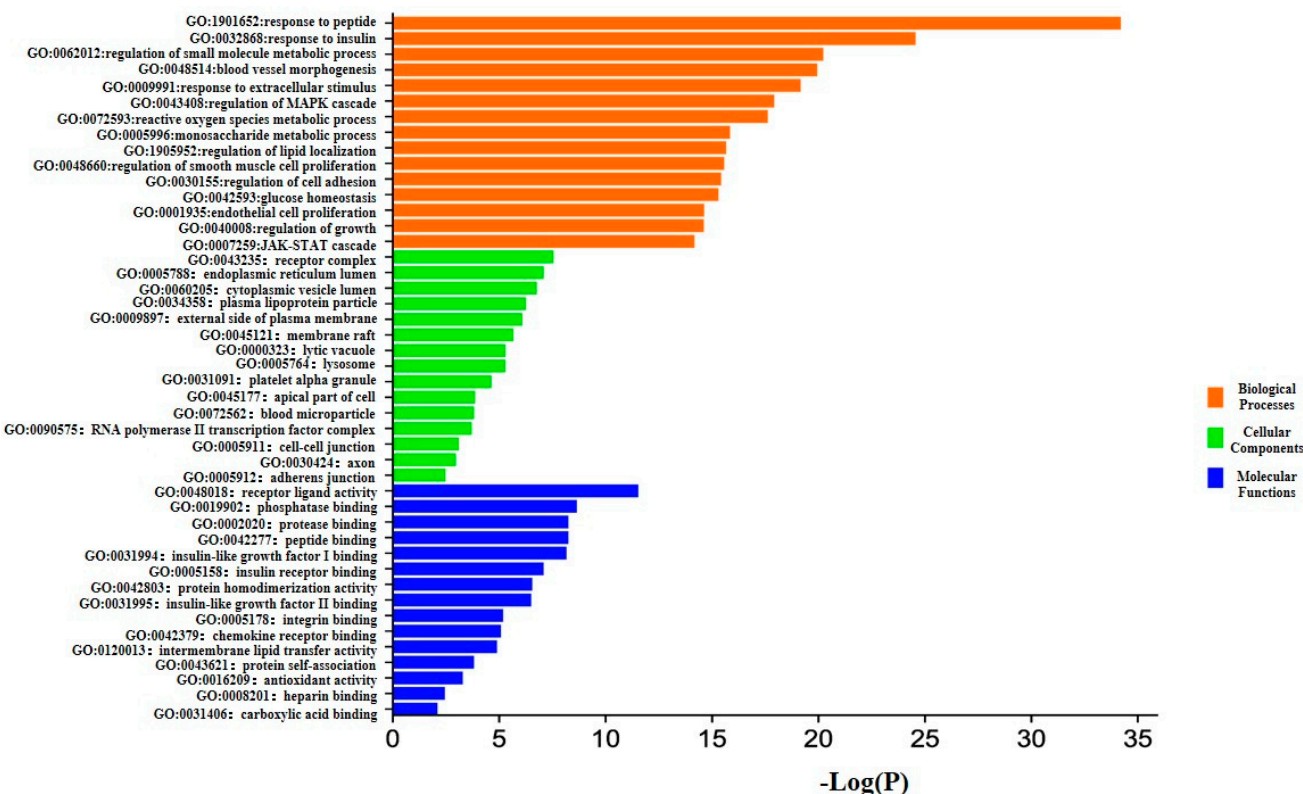

**Figure 3.** GO enrichment analysis of the target proteins. Biological Process (**Orange**), Molecular Function (**Blue**), and Cellular Component (**Green**).

KEGG pathway enrichment analysis was also performed by DAVID, and the top 25 pathways involving 51 target proteins were detected by BH-corrected *p*-values < 0.05 (Figure 4a). The results showed that these target proteins mainly participated in the AGE-RAGE signaling pathway in diabetic complications, insulin resistance, HIF-1 signaling pathway, FoxO signaling pathway, MAPK signaling pathway, and the PI3K-AKT signaling pathway. The AGE-RAGE signaling pathway, which exhibited the highest number of target connections, was an important segment in the occurrence of diabetic nephropathy, involving a variety of targets such as NF-κB, VEGF, and TGF-β1 [32–34]; both the insulin resistance and the HIF-1 signaling pathway were well-established heavily in insulin secretion and glucose homeostasis [35]. Besides, the PI3K-AKT signaling pathway is sensitized, and plays a key role in the regulation of glucose metabolism and the synthesis of glycogen and protein when the insulin receptors bind to the insulin receptor substrate [36]. The key targets and the major signaling pathway of the active ingredients of *Pueraria lobata* for DM and its complications are described in Figure 4b.

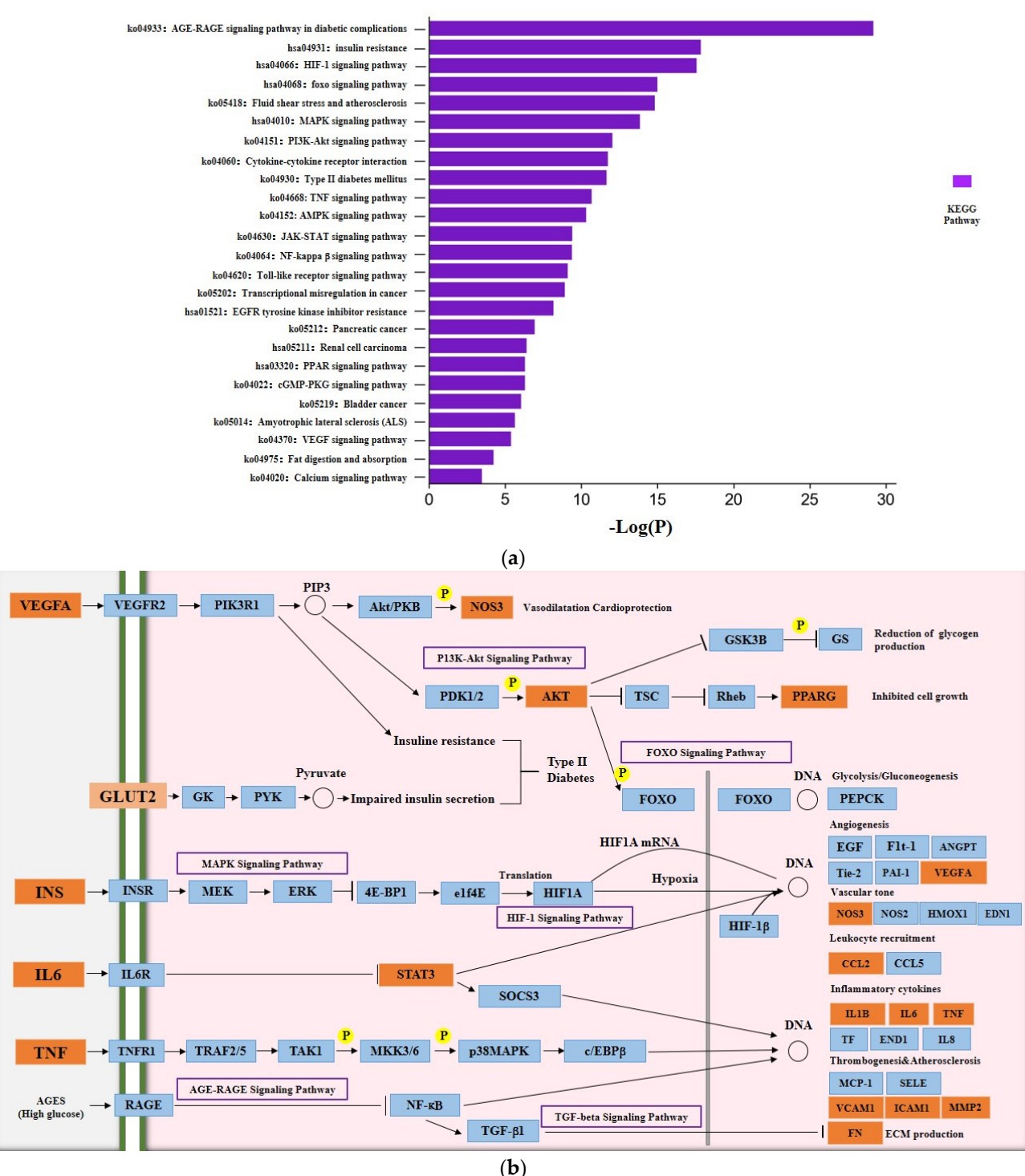

**Figure 4.** KEGG analysis of target proteins (**a**) and distribution of the target proteins of *Pueraria lobata* on the predicted pathway (**b**). The orange nodes are potential target proteins of *Pueraria lobata*, while the blue nodes are relevant targets in the pathway.

### 4.5. Network Construction and Analysis of DM-Related Ingredients

Recently, network pharmacology provides an effective tool for the study of TCM pharmacology and exerts a systematic method to analyze the mechanism of action of multi-target and multi-path drugs at the macro level. In this work, databases were mined including TCMSP, Uniprot, and Drugbank; Cytoscape 3.8.2 was used to analyze the interaction between 19 active ingredients, target proteins, and pathways. To decipher the

visualization of the complex relationship of the effective compounds of *Pueraria lobata* and their target proteins, a graph of the network was shown in Figure 5a. The results included a total of 337 nodes and 703 edges, of which nodes represented the targets of the 19 effective ingredients and the edges represented the interactions of the compounds and target proteins. All of the targets interacting with the active ingredients were mapped onto the top 25 KEGG pathways, and the disease–compounds–targets pathways network construction was generated. The result showed that puerarin, quercetin, genistein, daidzein, and other active ingredients in *Pueraria lobata* could participate in the AGE-RAGE signaling pathway, insulin resistance, HIF-1 signaling pathway, FoxO signaling pathway, and MAPK signaling pathway by acting on VEGFA, INS, INSR, IL-6, TNF and AKT1 (Figure 5b).

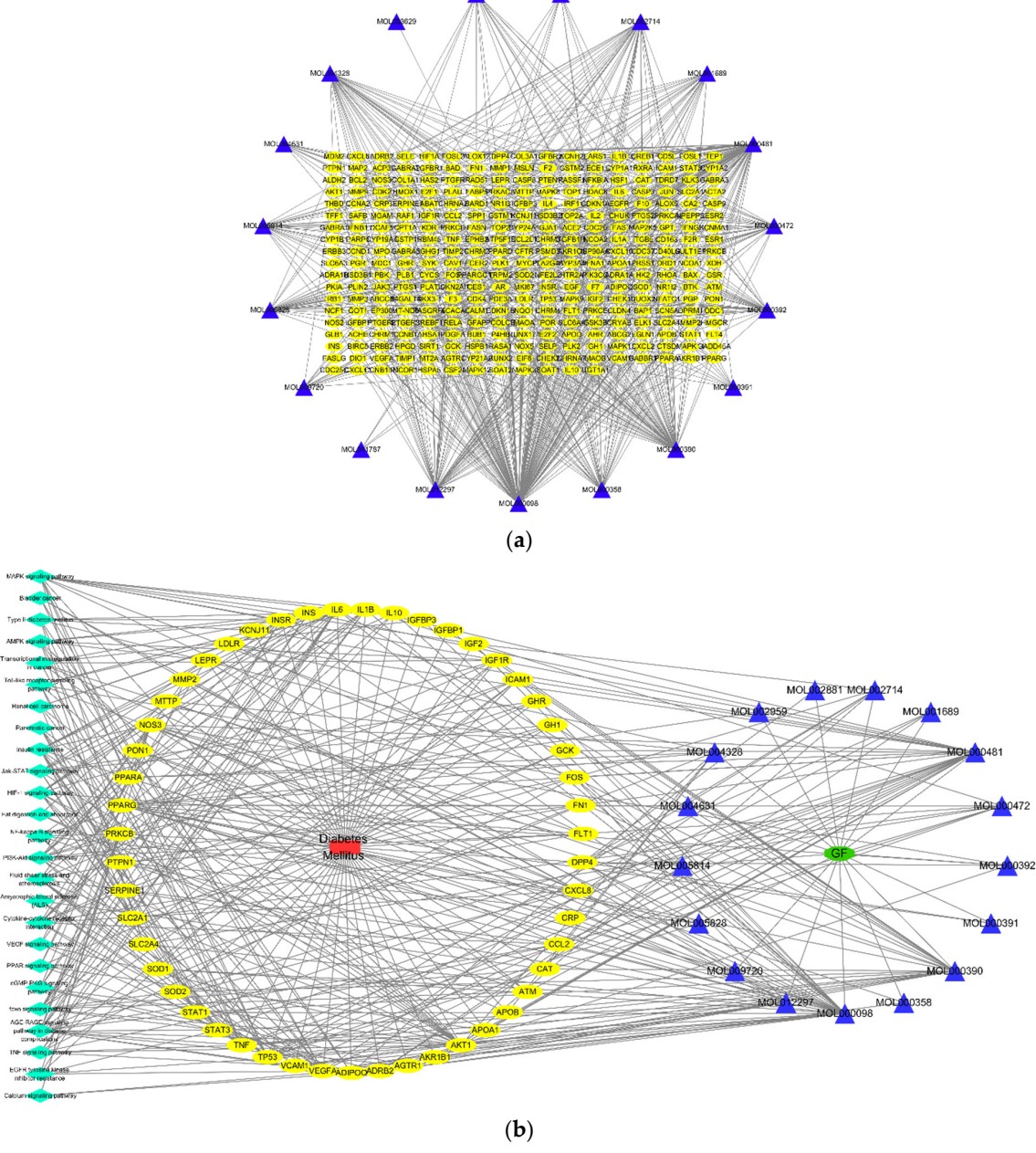

(**a**)

(**b**)

**Figure 5.** Network construction of "Ingredient-Target" (**a**). The blue triangles are the potential ingredients, while the yellow nodes are relevant targets of *Pueraria lobata*. Network construction of "Drug–Ingredient–Target Pathway" of *Pueraria lobata* (**b**). The green represents the drug and the blues are the ingredients, while the yellow nodes are relevant targets of DM, and the wathets are the possible pathways.

## 5. Discussion

DM can be defined as the syndrome of thirst elimination (the syndrome of Xiao Ke) in TCM, which is usually associated with Qi deficiency (Qi and blood) and Yin deficiency (body fluid). Thus, the "Gan" (sweet taste) Chinese Herbal Medicines are often used to relieve the syndrome [37–39]. Through network and data analysis of TCM prescriptions for DM, it is found that Chinese herbal medicines, characterized as tonifying Qi and Yin, which activate blood circulation to dissipate blood stasis, clear heat, and promote fluid are more commonly used. In terms of the application frequency, *Radix Astragali*, *Rhizoma Dioscoreae*, *Salviae Miltiorrhizae*, *Rehmanniae Radix*, and *Pueraria lobata* are the most commonly used of the TCM [40–42]. In our work, a complete system pharmacological approach was successfully applied to elucidate the molecular mechanism of *Pueraria lobata* in the intervention of DM. A total 19 active components and 51 corresponding DM-related targets were screened and selected, which mainly participated in 25 KEGG signaling pathways related to DM. Through systematic analysis, puerarin, quercetin, genistein, and daidzein of *Pueraria lobata* could mainly stimulate a majority of the specific procedures such as insulin secretion, improving insulin resistance, regulating the generation of glycogen and inflammatory cytokines, and improving endocrine metabolism, to achieve the effect of treating DM.

Long-term and persistent hyperglycemia increases the risk of cardiovascular, ocular, renal, and neurological diseases in diabetic patients. Among the complications, diabetic cardiomyopathy, diabetic nephropathy, and diabetic retinopathy are more common. The AGE-RAGE signaling pathway is considered to be the important step in the occurrence and development of diabetic nephropathy, and NF-kB, TGF $\beta$1, VEGF and MCP-1 are considered to be the key factors involved in the pathway and play a direct or indirect role [43–45]. Hypoxia-inducible factor HIF-1, including HIF-1$\alpha$ and HIF-1$\beta$, is an important factor widely present in the cell hypoxia environment, which plays diverse roles in different stages of diabetic nephropathy. In the early stage, HIF-1$\alpha$ can promote angiogenesis and improve the microcirculation of renal blood vessels by regulating EGF and VEGF. However, with the continuous progression of the disease, hyperglycemia inhibits the production of HIF-1$\alpha$ and increases the expression of ET-1, which leads to renal fibrosis. The protective effect of HIF-1 on the kidney is weakened, and the effect of cell apoptosis becomes visible [46–48]. FoxO1, a fork-head box transcription factor in the FoxO family, can cause oxidative stress, glucose and lipid metabolism disorders, inflammation, cell apoptosis, and other changes in cardiac structure and metabolism, thus playing an important role in the pathogenesis of diabetic cardiomyopathy [49,50]. VEGFA, a ligand of VEGFR2, is the most widely studied factor related to diabetic retinopathy. In the late stage of diabetic retinopathy, it can promote the proliferation of endothelial cells and participate in the pathological growth of new blood vessels [51–53]. Finally, in our present study, it was reasonable to believe that *Pueraria lobata* could improve the progression of diabetic cardiomyopathy, nephropathy, and retinopathy by interfering in MAPK, AGE-RAGE, HIF-1, and other signaling pathways through VEGFA, IL-6, MCP-1, FN, VCAM1, ICAM1, and NOS3. These findings were helpful to provide a theoretical basis for the treatment of DM and its complications with *Pueraria lobata*.

Among the signaling pathways associated with DM, insulin resistance plays an important role in the pathogenesis of DM. Insulin binding to the insulin receptor is the first step of signal transduction, while the active IRS protein can recruit and activate a variety of signal transduction proteins, which mediates the signal transduction effects such as IRS and IGF. Meanwhile, the PI3K signaling pathway exerts important effects on the pathogenesis of DM and its complications. PI3K, a lipid kinase, which can be activated by the regulatory subunit of PI3K (P85) binding to IRS, plays a key role in promoting the production of PIP, PIP2, or phosphatidylinositol triphosphate (PIP3), which are considered to be the second messengers of insulin and other growth factors. Particularly, PIP3, an important mediator of PI3K-dependent biological effects of insulin, directly binds to downstream signaling molecules of PI3K and regulates a variety of signal transductions

such as exposing phosphorylation sites, forming specific signal complex by promoting the target molecules to gather to the cell membrane and regulating the catalytic activity. Phosphoinositide 3-dependent kinase (PDK, both PDK1 and PDK2) and protein kinase B (PKB) make up downstream PI3K signaling molecules. PKB is a key molecule in the PI3K signaling pathway, and can be activated by the phosphorylation of PDK1 and PDK2. Furthermore, PKB can participate in a lot of biological procession such as glycogen and protein synthesis, inhibition of apoptosis, glucose transport, and mediate in the survival pathway of β cells, which is closely related to the growth, proliferation, and apoptosis of β cells [54,55]. The results of pathway enrichment in our study also indicated that *Pueraria lobata* could play a key role in the occurrence of DM and its complications, by intervening with the PI3K signal pathway and the insulin signal transduction pathway.

In conclusion, our study illuminated that the possible targets of the active ingredients of *Pueraria lobata* intervened in the related signaling pathways for DM. Preliminarily, we clarified the synergistic effects of ingredient-target-diabetes, which could provide the material basis of anti-diabetic and anti-inflammatory effects of *Pueraria lobata* as a hypoglycemic food.

**Supplementary Materials:** The following are available online at https://www.mdpi.com/article/10.3390/pr9071245/s1, Figure S1: the base peak chromatogram of positive and negative ions and the composition of the chemical metabolites, Figure S2: the common targets of ingredients and DM. The statistical bar chart of proteins with degree values greater than 12, Table S1: a total of 256 chemical ingredients of *Pueraria lobata*.

**Author Contributions:** S.Z. wrote the paper, S.Z. and Q.G. analyzed the data, K.C. designed the research. Resources, L.C. All authors have read and agreed to the published version of the manuscript.

**Funding:** This work was supported by the National Natural Science Foundation of China (grant numbers 31861143051, 31872425).

**Institutional Review Board Statement:** Not applicable.

**Informed Consent Statement:** Not applicable.

**Data Availability Statement:** Not applicable.

**Conflicts of Interest:** The authors declare that they have no known competing financial interests or personal relationships that could have appeared to influence the work reported in this paper.

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
