# Peer review of "Studies of the Anti-Diabetic Mechanism of Pueraria lobata Based on Metabolomics and Network Pharmacology"

_processes, doi:10.3390/pr9071245_

Round 1

Reviewer 1 Report

Manuscript entitled “Studies of the anti-Diabetic Mechanism of Pueraria lobata Based on Metabolomics and Network Pharmacology" is within the scope of Processes. It is a good experimental article with an interesting subject and good experimental work. The manuscript is fairly well written and includes a great deal of information. The results and discussion are represented in a logical way. However, the following suggestion should be carried out before the acceptance.

  1. Abstract: This statement was not clear "the potential molecular mechanism of Pueraria lobata on DM has not been distinctly revealed". Because there are several reports documented the possible anti-diabetic mechanism of Pueraria lobata.
  2. Introduction : The authors advised adding recent epidemiological data of Diabetes mellitus.
  3. Computer-based studies are enough to validate the anti-diabetic activity of Diabetes mellitus. If so possible, authors may be advised to include in vitro or in vivo studies.
  4. Methodology The plant material authentication was missing.
  5. The status of the plant leaves is not certain: is it yellow or green? what is the plant growth stages when you get the leaves? How did you keep the leaves before you carried out the study? All these factors have a great influence on the stability of your results.
  6. What is the major composition in your plant extract? And, how about their contents in the extract? Are they proteins, phenols, etc? All these will affect your results too.
  7. The results may be attractive if the author may incorporate Lipinski's rule or toxicity profile validation using ADMET-SAR online tool.
  8. Carefully check the abbreviation.
  9. Table 1: The quality of the chemical structure was flawed. ChemDraw images replaced it.
  10. Results: Based on the preliminary studies, 19 active components were selected for these studies. To my best of knowledge, most of the compounds pharmacological activities were already validated. So hoe the authors explore the novelty of the work.
  11. Discussion : Line No: 244-247. These sentences were confusing.
  12. Line No : 280. How the authors believe that Pueraria lobata could improve the progression of diabetic cardiomyopathy, nephropathy, and retinopathy. If so, which molecule responsible for its multi-faced pharmacological activity?
  13. The discussion party part is well established. Although in this section the Authors should add more details related to the anti-diabetic activities of the phytocompounds.https://doi.org/10.1016/j.fct.2021.112374;https://doi.org/10.3390/ph14020102
  14. https://doi.org/10.3390/antiox10010099
  15. References : It was advised to make the uniform citation of all the references.

Author Response

Dear Reviewer:

Thank you for your comments for our manuscript entitled “Studies of the anti-Diabetic Mechanism of Pueraria lobata Based on Metabolomics and Network Pharmacology”. Those comments are very helpful for improving our paper, as well as the good guiding significance for our later research. We have read the comments carefully and made revision which we hope meet with approval.

We will reply to your comments one by one.

  1. Thank you very much for pointing out and reminding that the expression here was inaccurate. We had revised this statement which was in Page 2 line 18-20 and please check it in the manuscript. Thank you very much.
  2. Here we cited the data of the global diabetes epidemiology published by the IDF on its official website. The data were updated for each two years, and the latest research results were expected to be released in the second half of 2021. In addition, we choosed the data of DM in China were the sixth large-scale epidemiological investigations conducted in China on DM, completed by a number of experts from endocrinology branch of the Chinese Medical Association and publish in BMJ. Although it was the data before and after 2019, we thought it was relatively comprehensive, and its credibility and authority were relatively high. So, we cited these datas.

However, we also realize that some expressions in the article were not standard enough. We had revised them, which were reflected in page 3 line 35-36 and line 40 of the manuscript. Thank you for your comments, please verify them in the manuscript.

  1. Your comments is very useful for us and we also agreed that further pharmacological studies could be carried out in vivo or in vitro. However, due to the reason of COVID-2019, the school laboratory had been closed for some time. We will select the important active components in Pueraria lobata in the future to conduct a separate experimental study. Thank you very much.
  2. For questions 4 and 5, we are very sorry that we did not explain the source and the production method of Pueraria clearly in the material method. We added it in the manuscript in Page 4 line 78-81. Please check.

We sprayed water on the fresh Pueraria lobata and wrapped it with plastic wrap and then stored in the refrigerator at 4℃ in our lab. The color of the powder after treatment was gray white.

  1. We had provided a supplementary material with the information about the compounds of Pueraria lobata identified by LC-MS. According to your comments, we had modified the supplementary material to supplement the classification of these ingredients. The information was also listed at the end of our reply for you.
  2. Thank you for your suggestion. ADMET-SAR online tool was a new knowledge and challenge for me. We will carefully study your proposal and use this technology in our future related research.
  3. I am very sorry that some abbreviations and semantic expressions in our manuscript were inaccurate. We checked them again. Thank you very much.
  4. We had redrawn the chemical formula by using the online drawing software, which had been shown in Table 1. Thank you very much.
  5. The 19 active substances we screened out are all reported substances with pharmacological activities. However, the innovation of this study is that we first used LC-MS metabonomics method to analyze and identify all the possible compounds of Pueraria lobata. On the basis of the identified components, we screened the active components with the existing network database of traditional Chinese Medicine, Finally, network pharmacology method was used to analyze the possible relationship between active ingredients and targets from the macro level, and systematically and comprehensively predict the effect of these active ingredients.

In the past many years, the single "component target" research mode has been playing a leading role in drug research and development. However, because the biological body is a complex structural network, and diseases are often associated with many genes, functional proteins and metabolic pathways, we want to use network pharmacology from the macro level, We first speculate or pre screen out the potential targets of drugs. Of course, in the future, we will conduct in-depth research on certain active ingredients and certain targets in combination with in vivo and in vitro experiments.

  1. We had revised the statements in line 250-252.
  2. According to the official website of IDF, long-term and continuous hyperglycemia will increase the risk of cardiovascular, eye, kidney and neurological diseases in diabetic patients. And Diabetic retinopathy, diabetic neuropathy and oral complications are common. Many studies have mentioned that HIF-1, FoxO1, VEGFA and other key factors can play a role in diabetic nephropathy, cardiomyopathy and retinopathy. Our study suggests that these active substances in Pueraria lobata can be involved in the process of controlling the complications of diabetes through VEGFA, IL6, MCP-1, FN, VCAM1, ICAM1, NOS3 and other signal pathways, such as MAPK, AGE-RAGE and HIF-1. Of course, this is a prediction. In the future, we will further study in vivo and in vitro by selecting these key factors.
  3. We had read and added these references in the discussion in line 233 in our manuscript.
  4. We revised the format of the references. Thank you very much.

Reviewer 2 Report

The paper entitled “Studies of the anti-Diabetic Mechanism of Pueraria lobata Based on Metabolomics and Network Pharmacology” by Shu Zhang, Qi Ge, Liang Chen and Keping Chen showed how possible targets of active compounds from Pueraria lobata intervened in signaling pathways related to Diabetes mellitus.

The study is well articulated.  The authors are very conversant with the area and methodology. However, the following corrections must be made:

- page 2 (lines 42 and 45), page 4 line 107: references are repeated several times.

- page 5, line 114: Change “relevancescore>28” to “relevance score > 28”.

- table 1: the numbers in the molecular formula are subscribed.

- the table 1 needs to be reorganized so that the structures can be increased. As presented, they are illegible.

- tables must be formatted correctly, keeping only the top and bottom rows.

- Supplementary Table S1: the numbers in the molecular formula are subscribed.

- the name 5′-Deoxy-5′-<WBR>(methylthio)adenosine is correct?

Author Response

Dear Reviewer:

Thank you for your comments for our manuscript entitled “Studies of the anti-Diabetic Mechanism of Pueraria lobata Based on Metabolomics and Network Pharmacology”. Those comments are very helpful for improving our paper, as well as the good guiding significance for our later research. We have read the comments carefully and made revision which we hope meet with approval.

We will reply to your comments one by one.

  1. I'm very sorry, there are some mistakes in our references, which had been revised. Thank you very much.
  2. We had changed “relevancescore>28” to “relevance score > 28”. Line 123 and 176
  3. According to your comments, we had adjusted the format of the chemical formula.
  4. We had redrawn the chemical formula by using the online drawing software, which had been shown in Table 1. Thank you very much.
  5. We reformatted the table.
  6. According to your comments, we had modified the supplementary material to supplement the classification of these ingredients. The information was also listed at the end of our reply for you. Thank you very much.
